# Dynamic diversity of synthetic supramolecular polymers in water as revealed by hydrogen/deuterium exchange

Xianwen Lou[1],*, René P.M. Lafleur[1],*, Christianus M.A. Leenders[1], Sandra M.C. Schoenmakers[1], Nicholas M. Matsumoto[1], Matthew B. Baker[1], Joost L.J. van Dongen[1], Anja R.A. Palmans[1] & E.W. Meijer[1]

Numerous self-assembling molecules have been synthesized aiming at mimicking both the structural and dynamic properties found in living systems. Here we show the application of hydrogen/deuterium exchange (HDX) mass spectrometry (MS) to unravel the nanoscale organization and the structural dynamics of synthetic supramolecular polymers in water. We select benzene-1,3,5-tricarboxamide (BTA) derivatives that self-assemble in $H_2O$ to illustrate the strength of this technique for supramolecular polymers. The BTA structure has six exchangeable hydrogen atoms and we follow their exchange as a function of time after diluting the $H_2O$ solution with a 100-fold excess of $D_2O$. The kinetic H/D exchange profiles reveal that these supramolecular polymers in water are dynamically diverse; a notion that has previously not been observed using other techniques. In addition, we report that small changes in the molecular structure can be used to control the dynamics of synthetic supramolecular polymers in water.

[1] Institute for Complex Molecular Systems, Eindhoven University of Technology, PO Box 513, Eindhoven 5600 MB, The Netherlands. * These authors contributed equally to this work. Correspondence and requests for materials should be addressed to A.R.A.P. (email: a.palmans@tue.nl) or to E.W.M. (email: e.w.meijer@tue.nl).

One-dimensional (1D) supramolecular polymers in water represent an interesting class of materials, because their inherent dynamic behaviour can be tuned to match the dynamic behaviour of the supramolecular interactions found in living tissue[1]. Since these dynamic interactions are essential for many cell functions, supramolecular polymers and gels based on these polymers are a great platform en-route to biomedical applications[2]. Several supramolecular building blocks form 1D aggregates in water, for example, peptide-amphiphiles[3–10]. Recent work by us and others highlight the importance of kinetic considerations next to thermodynamic considerations; the nature of the architectures formed is controlled by the supramolecular pathway selected[11–15]. Time-resolved measurements are of key importance to unravel the mechanisms by which molecules self-assemble[16,17] and can provide insight into the internal organization of supramolecular fibres. Recently, supramolecular peptide-based fibres containing spin labels were shown to exhibit a range of dynamics within different domains along the cross-section of the fibres[18]. Despite the interesting heterogeneity in the motions observed, introducing molecular probes into supramolecular aggregates will without doubt have an influence on the local intermolecular interactions and hence the dynamic behaviour. Experimental measurements addressing the motion of molecules, that do not require molecular probes, can therefore provide unique structural insights.

Hydrogen/deuterium exchange (HDX) mass spectrometry (MS) is a well-established technique in biochemistry to study a variety of systems, including the structure, dynamics and folding of proteins[19–26]. HDX-MS does not require the introduction of a molecular probe. In deuterated water, amide hydrogens that are not involved in hydrogen bonding and are solvent accessible undergo rapid amide H/D exchange, increasing the molecular weight of the protein by one Dalton per exchange event. Other amide hydrogen atoms exchange much slower or not at all. HDX-MS has afforded detailed insight into the mechanism of aggregation of proteins into amyloid fibres, and the technique also revealed that the proteins are in a continuous dynamic equilibrium between fibres and the molecularly dissolved state[27–29]. A first example of the use of this technique on synthetic supramolecular structures in the gas phase was recently published by Schalley et al[30].

The work of Schalley and the beautiful work performed on elucidating the mechanism of amyloid fibre aggregation by HDX (refs 27–30) indicate that HDX-MS can also be applied as a label-free method to elucidate the dynamic processes involved in synthetic supramolecular self-assemblies in water. In order to assess the applicability of HDX-MS for supramolecular self-assembled structures in water, we selected our well-studied benzene-1,3,5-tricarboxamide (BTA) motif[31]. This BTA derivative has been studied in detail with different techniques, while still many questions concerning the molecular dynamics remained unanswered[32–34]. $C_{12}BTA$ self-assembles in water via a combination of hydrophobic interactions and directional hydrogen-bond formation into long, supramolecular polymers[31]. Dodecyl spacers shield the hydrogen bonds from water and a hydrophilic tetra(ethylene) glycol is attached to the aliphatic spacers, surrounding the hydrophobic pocket, to ensure water compatibility. The association constant responsible for the self-assembly of $C_{12}BTA$ is so high that we cannot measure a critical association concentration of the cylindrical and/or spherical aggregates yet. This is in line with the very large aspect ratio of these supramolecular polymers. Molecular dynamics (all-atom MD) simulations on $C_{12}BTA$ in water revealed the molecular fluctuations of the supramolecular polymer[32]. Previous experimental investigations focused on dye-labelled variants of $C_{12}BTA$ co-assembled with $C_{12}BTA$.

The formed supramolecular fibres were investigated by Förster resonance energy transfer (FRET)[33] and stochastic optical reconstruction microscopy (STORM)[34], which revealed that the dye-labelled monomers exchange from fibre-to-fibre by a release-incorporation mechanism. This monomer exchange showed half-lives in the order of hours at room temperature and appeared to occur randomly, within the 50 nanometre resolution of the STORM technique, along the entire supramolecular polymers[34].

In this contribution, we show that HDX-MS is a promising technique to study the dynamics of the water-soluble supramolecular polymer based on the BTA motif, that is, the physical movements of the constituent monomers, without the use of molecular probes. In the first part, we show the H/D exchange of supramolecular polymers formed from $C_{12}BTA$ and validate the method of HDX-MS. In the second part of the paper, time-resolved measurements reveal, to our surprise, that the HDX of BTA polymers in water is governed by different processes that occur at different timescales. In addition, control over the polymer dynamics was achieved by varying the temperature and molecular structure. Our results show that HDX-MS is a powerful method for the investigation of synthetic supramolecular self-assembly in water, and the different timescales observed suggest that the supramolecular polymers studied here are structurally diverse.

## Results

**The initial H/D exchange experiment**. The BTA-based supramolecular polymers formed from $C_{12}BTA$ are made in $H_2O$ using a non-covalent synthetic protocol by heating, vortexing, followed by slow cooling. Once formed, the 500 μM $C_{12}BTA$-$H_2O$ solution is diluted 100 times into $D_2O$, after which mass spectra are taken in time, by using electrospray ionization mass spectrometry (ESI-MS). Surprisingly, only two major isotopic distributions corresponding to $C_{12}BTA3D$ and $C_{12}BTA6D$ are observed after 1 h. The contribution of $C_{12}BTA6D$ increases over time. The $C_{12}BTA$ unit has three hydroxyl hydrogens at the tetra(ethylene glycol) peripheries and three amide hydrogens at the core (Fig. 1). Indeed, a maximum of six hydrogen atoms are able to undergo H→D exchange (HDX) reactions with the surrounding solvent. The exchange rate of OH and amide NH depends on the solvent accessibility to these exchangeable hydrogens[24], because the acidity (pKa) of both OH and amide-NH is similar in water (pKa around 14.2 for OH and around 13.0 for NH, see Supplementary Information). The three OH groups of the hydrophilic tetra(ethylene glycol) motifs exchange immediately to OD as they are exposed to the surrounding aqueous medium, resulting in $C_{12}BTA3D$ (Fig. 1). For the NH groups, on the other hand, the exchange rate is much slower due to the reduced water accessibility, caused by the formation of the hydrophobic pocket. Remarkably, some of the amide exchange of NH to ND occurs at a timescale of hours, whereas other amides are not deuterated even after multiple days. Before going into the details of these interesting time-resolved observations, we will first validate the method of HDX-MS for supramolecular polymers.

**Validation of HDX-MS**. To validate the method, we assessed whether the isotopic distributions recorded with MS are representative of the real isotopic distributions in solution for the supramolecular polymers based on $C_{12}BTA$. There is the possibility that H/D exchange takes place during electrospray ionization (ESI). A viable way to validate the HDX-MS method is to find the conditions necessary to completely quench the H/D exchange. If the exchange is completely quenched in solution,

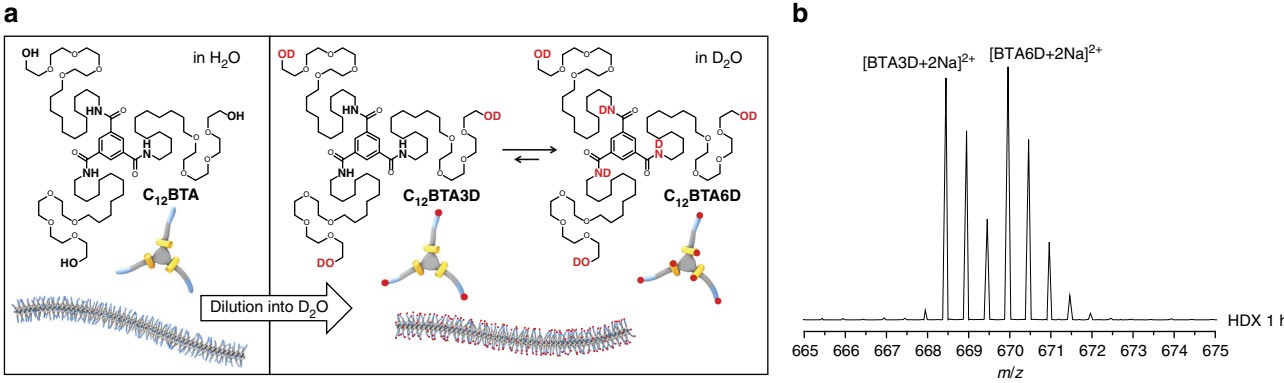

**Figure 1 | The H→D exchange reaction.** (**a**) Chemical structures of C$_{12}$BTA and its deuterated analogues C$_{12}$BTA3D and C$_{12}$BTA6D, schematic representation of the supramolecular polymers formed in water and an illustration of the H/D exchange process. (**b**) ESI-MS spectrum of C$_{12}$BTA diluted into D$_2$O after 1h of exchange.

the contribution of the HDX that occurs in the ESI chamber can be determined. A number of methods described in the literature (including: extraction with a dry aprotic organic solvent, cooling down to 0 °C at pH = 2.3 and freeze-drying to remove H$_2$O/D$_2$O followed by re-dissolving in a dry aprotic organic solvent[35,36]) were evaluated for their ability to quench the exchange. Under these conditions, the exchange was never quenched effectively. However, at 0 °C, a temperature at which the exchange of dye-labelled monomers between different supramolecular polymers in solution is significantly slowed down[33], less than 7% of C$_{12}$BTA6D was obtained after HDX for 1 min (Supplementary Tables 1,2) by diluting C$_{12}$BTA-H$_2$O 100 times in D$_2$O. This value is in the same range as observed for proteins, which is typically <4% (ref. 25). Thus, despite the difficulty of finding the experimental conditions required to completely quench the H/D exchange, and considering the relatively fast exchange rate directly after dilution of a C$_{12}$BTA-H$_2$O solution into D$_2$O (see below), we conclude that the contribution of H/D exchange in the ESI chamber is limited for our synthetic supramolecular polymers.

In the ESI-MS measurements of the C$_{12}$BTA polymers in water, peaks of singly, doubly and triply charged sodium adducts of the BTA molecules were observed, among which the doubly charged ion was the most pronounced species (Fig. 2a). Although we observed a large difference in the signal intensity in a given mass spectrum, no difference in the respective isotopic distributions was observed for the ions with different charges. Therefore, the doubly charged ions were used for probing HDX of the supramolecular polymers. We never observed masses corresponding to dimers or lower oligomers in the mass spectra, indicating that full depolymerization of the supramolecular polymers occurred as a consequence of the ionization settings we employed.

A reference mass spectrum of a C$_{12}$BTA sample without deuterium exchange was measured by the dilution of a 500 µM C$_{12}$BTA-H$_2$O solution 100 times into H$_2$O (Fig. 2b, spectrum in green). The isotopic pattern due to the naturally occurring isotopes of the composing elements is observed in this reference spectrum. On the contrary, after the dilution of a 500 µM C$_{12}$BTA-H$_2$O 100 times into D$_2$O and HDX for 1 h, two major isotopic distributions corresponding to [C$_{12}$BTA3D + 2Na]$^{2+}$ and [C$_{12}$BTA6D + 2Na]$^{2+}$ were observed (Fig. 2b, black spectrum). The isotopic pattern for [C$_{12}$BTA3D + 2Na]$^{2+}$ is the result of the rapid exchange of the hydrogens of the OH groups for deuterium, and the pattern for [C$_{12}$BTA6D + 2Na]$^{2+}$ is the result of HDX of both the OH and the NH groups. By

calculating and comparing the relative peak intensities in the isotopic distributions, it was observed that the relative intensity of the first two neighbouring isotopic peaks for C$_{12}$BTA3D ($m/z$ at 668.46 and 668.96, Fig. 2c) was the same as that for the non-deuterated BTA counterpart (C$_{12}$BTA, $m/z$ at 666.95 and 667.45, green spectrum). This indicates that indeed no C$_{12}$BTA4D was present after HDX for 1 h. In the presence of C$_{12}$BTA4D, the relative intensity of the second isotopic peak ($m/z$ at 668.96) has to be significantly higher. A similar analysis indicates that C$_{12}$BTA5D was absent when the contribution of the trace amount of H$_2$O in the sample to the formation of C$_{12}$BTA5D was subtracted. The isotopic patterns observed, therefore, correspond to two distinct isotopic distributions of C$_{12}$BTA3D and C$_{12}$BTA6D without considerable contribution of C$_{12}$BTA4D and C$_{12}$BTA5D (Fig. 2b, spectra in black and red).

The intensity of the isotopic pattern for [C$_{12}$BTA6D + 2Na]$^{2+}$ increased with exchange time (Fig. 2b, red spectrum for HDX after 24 h, Supplementary Table 3). When calculating the percentage of C$_{12}$BTA3D, the overlap of the isotopic peaks of C$_{12}$BTA3D and C$_{12}$BTA6D should be taken into account. After 1 h of HDX, around 50% of C$_{12}$BTA3D was transformed to C$_{12}$BTA6D. After 24 h about 25% of C$_{12}$BTA3D remained, while still no 4D or 5D species were observed. As a control experiment, a 500 µM BTA solution in H$_2$O was diluted 100 times into acetonitrile/D$_2$O (1/1 v/v), since acetonitrile is known to dissolve BTAs (ref. 37). All C$_{12}$BTA3D was immediately transformed to C$_{12}$BTA6D (Fig. 2b, blue spectrum). The tiny peak in front of C$_{12}$BTA6D is due to the presence of a trace amount of the original H$_2$O.

In the experiments discussed above, BTA polymers prepared in H$_2$O were exposed to D$_2$O leading to the replacement of hydrogen (H) by deuterium (D) in an exchange-in scenario. To investigate whether the deuterium labels would affect the strength of the intermolecular hydrogen bonds and, as a result, the HDX rate of BTA molecules in the polymers, experiments were also performed in a reversed scenario of exchange-out. In this case, BTA-based supramolecular polymers were prepared in D$_2$O and fully deuterated before they were diluted into H$_2$O. No considerable influence of the D-labels on the HDX behaviour was observed. Similarly, the exchange rate in the 'exchange-out' mode was fast in the beginning with about 50% exchanged after 1 h, and slowed down significantly afterwards with about 25% not exchanged after 24 h (Supplementary Fig. 1). This result indicates that the possible difference in the strength of the ND···OC and NH···OC hydrogen bonds is not a factor that contributes to the observed decrease in the HDX rate over time.

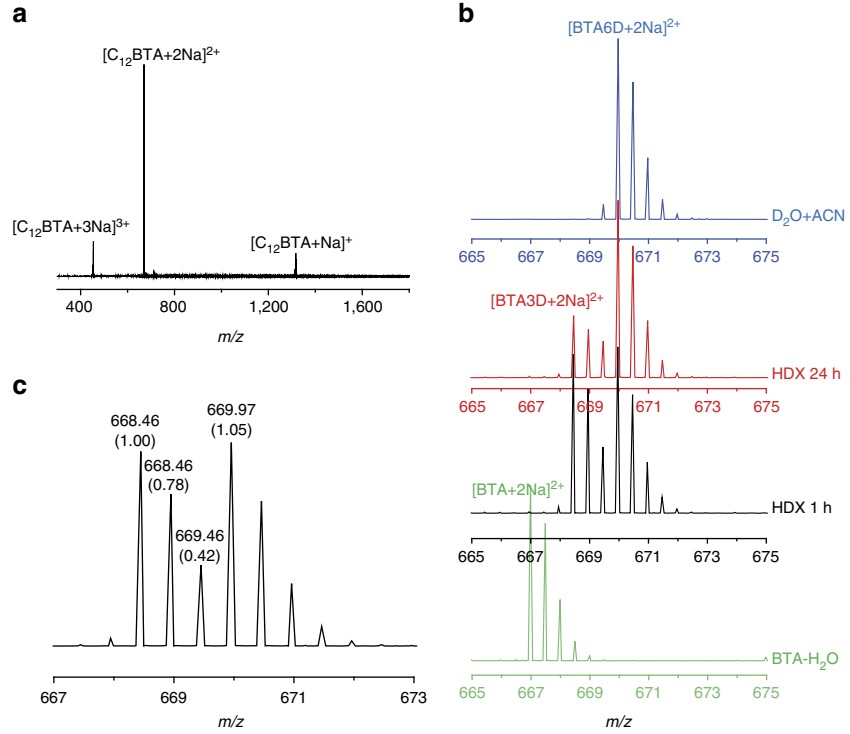

**Figure 2 | Mass spectra during the exchange reaction.** (**a**) ESI-MS spectrum of $C_{12}BTA$ with the doubly charged ion as the most prominent ion. (**b**) HDX-MS spectra of doubly charged sodium adducts of $C_{12}BTA$ obtained after the dilution of a 500 µM $C_{12}BTA$-$H_2O$ solution 100 times into $D_2O$ or $H_2O$. The spectra at four different stages are presented in different colours. The green spectrum is a reference spectrum of $C_{12}BTA$ diluted into $H_2O$; the black spectrum and the red spectrum are recorded after HDX for 1 and 24 h, respectively; the blue spectrum is the spectrum of a control measurement where the polymers were diluted 100 times into a solvent mixture containing acetonitrile (ACN/$D_2O$ = 1/1 (v/v)). (**c**) Mass spectrum obtained after the dilution of a 500 µM $C_{12}BTA$-$H_2O$ solution into $D_2O$ and HDX for 1h. The numbers in the round brackets are the corresponding relative intensities.

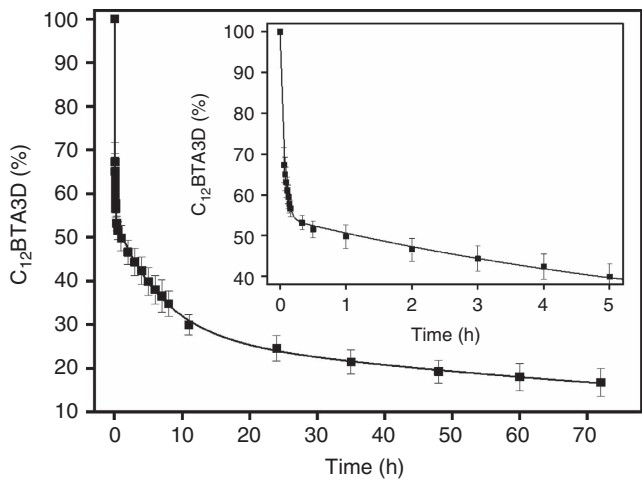

**Figure 3 | Time-dependent exchange experiment.** Percentage of $C_{12}BTA3D$ as a function of time after the dilution of a 500 µM BTA solution 100 times into $D_2O$, at room temperature. The black line is a tri-exponential fit of the experimental data including the point ($t = 0$ h, $C_{12}BTA3D = 100\%$), with $k_{initial} = 1.8 \times 10^1\,h^{-1}$, $k_{fast} = 1.4 \times 10^{-1}\,h^{-1}$ and $k_{slow} = 0.7 \times 10^{-2}\,h^{-1}$. The error bars represent one s.d. of uncertainty computed from three separate kinetic experiments. The inset shows an enlargement of the data points and fit for the first 5 h.

**Time-resolved H/D exchange of $C_{12}BTA$ supramolecular polymer.** The significantly reduced exchange rate with time prompted us to analyse the dynamics in more detail and at longer timescales. A kinetic plot of the percentage of BTA3D versus exchange

time was made and displayed in Fig. 3. In this plot, we included the hypothetical data point ($t = 0$ h, BTA3D = 100%). Hence, we assume that all OHs are instantaneously replaced by ODs upon dilution into $D_2O$, and as a result [$C_{12}BTA3D$] = 100%. The inset of Fig. 3 shows that at the first measurement point ($t = 3.5$ min), around 30% of $C_{12}BTA3D$ was converted into $C_{12}BTA6D$. The H/D exchange was very fast in the first 20 min, slowed down after 1 h and slowed down even more 10 h after dilution into $D_2O$. Remarkably, even after 70 h the H/D exchange from $C_{12}BTA3D$ to $C_{12}BTA6D$ was not complete and around 17% of $C_{12}BTA3D$ remained. Both a mono-exponential fit and a bi-exponential fit were inadequate to fit the data points. A tri-exponential fit, in contrast, fits the data well suggesting that the exchange process occurs with three different rates of exchange, with rate constants $k_{initial}$ of $1.8 \times 10^1$, $k_{fast}$ $1.4 \times 10^{-1}$ and $k_{slow}$ $0.7 \times 10^{-2}\,h^{-1}$. Interestingly, the data points after 1 h fit well to a bi-exponential decay with rate constants of $1.3 \times 10^{-1}$ and $0.7 \times 10^{-2}\,h^{-1}$, which are almost exactly the same as the corresponding values from the tri-exponential fit (Supplementary Fig. 2).

**H/D exchange as a function of temperature.** The HDX behaviour of the supramolecular polymers was studied in more detail at different temperatures. After the dilution of a 500 µM solution 100 times into $D_2O$, the diluted solutions were stored at 40, 50 and 55 °C and periodically subjected to ESI-MS starting 1 h after the dilution step. The results are summarized in Fig. 4; the measurements performed at room temperature are shown as a reference. The amount of $C_{12}BTA3D$ decreased rapidly with increasing temperature, hence the H/D exchange becomes faster with increasing temperature. The decrease of $C_{12}BTA3D$ over

time after 1 h fits in all cases well to a bi-exponential decay (Table 1, Supplementary Fig. 3). From Table 1 we can infer that both $k_{fast}$ and $k_{slow}$ increased with increasing the temperature, and that the contribution to HDX of the fast exchanging part increased at the expense of the slower exchanging part. Constructing van't Hoff plots based on these results was not successful, most probably due to the interconnection between both events. Whereas the kinetic HDX profiles recorded in the course of days for samples equilibrated at ambient temperature are highly reproducible, we found that kinetic profiles that were recorded at room temperature in different seasons showed

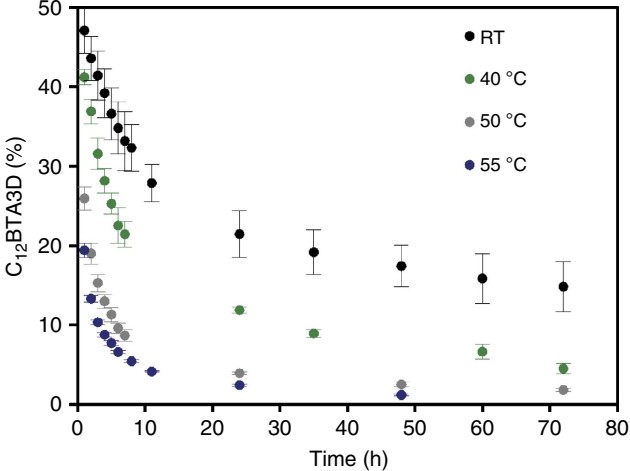

**Figure 4 | Temperature-dependent exchange kinetics.** Percentage of $C_{12}BTA3D$ over time when equilibrated at room temperature (black circles), 40 °C (green circles), 50 °C (grey circles) and 55 °C (blue circles). The error bars represent one s.d. of uncertainty computed from three separate kinetic experiments.

**Table 1 | Rate constants of HDX for the fast and slow exchanging parts together with the contributions of the fast and slow process at different temperatures (RT = room temperature).**

| Temperature | $k_{fast}$ (h$^{-1}$) | $k_{slow}$ (h$^{-1}$) | Fast (%) | Slow (%) |
|---|---|---|---|---|
| RT | $1.3 \times 10^{-1}$ | $0.7 \times 10^{-2}$ | 52.9 | 47.0 |
| 40 °C | $2.4 \times 10^{-1}$ | $1.7 \times 10^{-2}$ | 65.1 | 34.9 |
| 50 °C | $3.9 \times 10^{-1}$ | $2.4 \times 10^{-2}$ | 76.5 | 23.5 |
| 55 °C | $5.7 \times 10^{-1}$ | $4.8 \times 10^{-2}$ | * | * |

*Less than 8% of $C_{12}BTA3D$ was left after 5 h at 55 °C, hence calculation of these values was omitted.

slightly larger differences, especially in the amount of $C_{12}BTA3D$ present after 3 days of HDX. This indicates that the kinetic profiles of H/D exchange are highly sensitive to the exact temperature at which the solutions are equilibrated in $H_2O$ and stored after dilution into $D_2O$. No considerable differences in the HDX kinetics at ambient temperature were observed when the BTA polymers in $H_2O$ were prepared at 100 μM or 1 mM (Supplementary Table 4).

**Effect of the molecular structure on H/D exchange kinetics.** Only amide hydrogens that are not involved in hydrogen bonding and are solvent accessible can undergo H/D exchange. The solvent accessibility to the buried amide moieties within the supramolecular fibres depends on the size of the hydrophobic pocket. Previous results have shown that an undecyl spacer is the minimum length required to sufficiently shield the hydrogen bonds from water and form supramolecular polymers. BTAs with a decyl spacer form small disordered aggregates that are not stabilized by hydrogen bonds ($C_{10}BTA$ and $C_{11}BTA$, Fig. 5a)[38]. A BTA with a tridecyl spacer was synthesized to increase the size of the hydrophobic pocket ($C_{13}BTA$, Fig. 5a, Supplementary Methods and Supplementary Figs 4–7). Also this BTA was found to form supramolecular polymers in water (Supplementary Fig. 8), although the sample preparation is more complicated as compared to the other BTAs.

Also 500 μM solutions of the $C_{10}BTA$ and $C_{11}BTA$ were prepared using the same method as for $C_{12}BTA$; the sample preparation for $C_{13}BTA$ is discussed in the Methods Section. The H/D exchange experiments, performed at ambient temperature, started by the dilution of the BTA samples 100 times into $D_2O$. The results of the HDX measurements are presented in Fig. 5b. Interestingly, all exchangeable hydrogen atoms in $C_{10}BTA$ were fully exchanged to $C_{10}BTA6D$ at the first measuring point ($t = 3$ min). The H/D exchange profile for $C_{11}BTA$ was similar to $C_{12}BTA$ albeit that $C_{11}BTA3D$ converted faster into $C_{11}BTA6D$ as compared to the $C_{12}BTA$; after 48 h, 18% of $C_{11}BTA3D$ remained, whereas for $C_{12}BTA3D$ 29% remained. In contrast, $C_{13}BTA3D$ converted slower into $C_{13}BTA6D$ as compared to the $C_{12}BTA$; after 48 h, 44% of $C_{13}BTA3D$ remained. These differences in the conversion to the fully deuterated species are reflected in the rate constants $k_{slow}$ that are $1.5 \times 10^{-2}$ h$^{-1}$, $0.9 \times 10^{-2}$ h$^{-1}$ and $0.3 \times 10^{-2}$ h$^{-1}$ for increasing the size of the aliphatic spacer, respectively. None of these samples showed the presence of 4D and 5D species. These results confirm the value of HDX-MS as a suitable technique to obtain kinetic information on synthetic supramolecular aggregates in water, since differences in the nature of the self-assembled structures are clearly reflected by differences in the decay signature of the 3D to 6D species. By increasing the hydrophobic pocket, the proposed

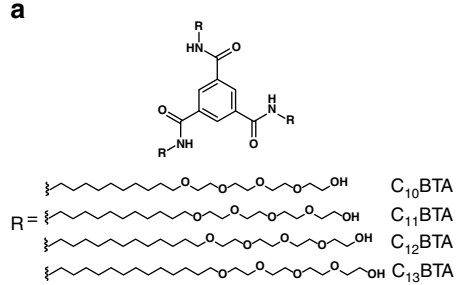

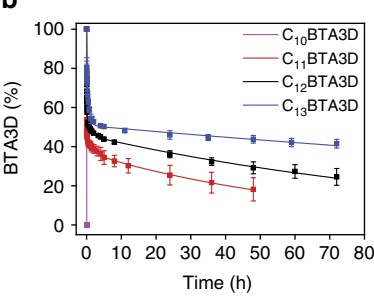

**Figure 5 | Structure-dependent exchange kinetics.** (**a**) Molecular structure of $C_{10}BTA$, $C_{11}BTA$, $C_{12}BTA$ and $C_{13}BTA$. (**b**) Corresponding percentages of BTA3D as a function of time after the dilution of the 500 μM BTA solutions 100 times into $D_2O$. Lines are tri-exponential fits of the data, and the error bars represent one s.d. of uncertainty computed from three separate kinetic experiments.

hydrogen bonds become more shielded from water. In order to get experimental evidence for a highly ordered hydrogen-bonded supramolecular polymer, we synthesized a chiral version of $C_{12}BTA$.

**CD reveals the emergence of structured domains**. Both solvent accessibility and the presence of hydrogen bonds influence the probability of H/D exchange of the amides. Local differences in the contributions of these two parameters along the polymer backbone could explain the different timescales observed within the H/D exchange profiles. To experimentally verify whether hydrogen bonds can influence the structural organization of the polymers, we investigated the degree of order within aggregates of $C_{12}BTA$ by preparing $(S)$-D-$C_{12}BTA$, a chiral, non-racemic BTA, in which a deuterium isotope was stereoselectively introduced at the α-position of all dodecyl spacers (Fig. 6a, Supplementary Methods, Supplementary Figs 9 and 10). The circular dichroism (CD) spectrum of $(S)$-D-$C_{12}BTA$ shows a remarkably strong Cotton effect (Fig. 6b). In BTA-based systems, this optical activity is directly related to the helical arrangement of the intermolecular NH···OC hydrogen bonds along the supramolecular polymer[31,38]. On top of that, we observed that the intensity of the CD spectrum increased over time and decreased after elevating the temperature (Supplementary Figs 11 and 12). Hence, we propose that supramolecular polymers of $(S)$-D-$C_{12}BTA$—and therefore also of $C_{12}BTA$—can build hydrogen-bonded helical domains that restrict the movement of the constituent monomers in the polymers, while their dynamics is dependent on the history of the sample.

## Discussion

HDX-MS is an important and well-established technique to study the non-covalent structural transitions of proteins in water. Whereas the conversion of NH to ND for exposed amides has a rate constant ($k_{HDX}$) in the order of seconds at neutral pH, the secondary and tertiary structures of proteins lower the observed rates due to the presence of hydrogen bonds and limited solvent access. As a consequence, also the rate constants of the structural fluctuations ('$k_{open}$' and '$k_{closed}$' for an open and closed structure, respectively) determine whether isotope exchange occurs at a specific amide. Under denaturing conditions $k_{HDX} \gg k_{closed}$, presenting a kinetic limit (referred to as EX1) where H/D exchange is very fast[23,26]. The other kinetic limit is characterized by $k_{closed} \gg k_{HDX}$, and referred to as EX2. Like for proteins at physiological conditions, the EX2 regime is more prevalent for our synthetic polymers since the hydrophobic effect and hydrogen bonds constitute the basis for the existence of these polymers. The EX1 regime is maybe more relevant when the polymers are diluted into acetonitrile/$D_2O$ (1/1 v/v), which is known to disrupt the polymers. However, it is important to note that the differences between biomacromolecules and supramolecular polymers are larger than the similarities. Despite these differences, the continuous labelling HDX experiments presented here are ideally suited to monitor the structural dynamics of supramolecular polymers.

Time-resolved HDX experiments on $C_{12}BTA$ indicate that the exchange occurs with three different rates of exchange. We propose three possible reasons for the very fast exchange observed in the first hour. Firstly, the very fast exchange could be due to the presence of small micelles that are not stabilized by intermolecular hydrogen bonds. Secondly, the 100-fold dilution step might shed weakly incorporated monomers from the polymer, or, thirdly, temporarily allow the penetration of $D_2O$ into the hydrophobic pocket during re-equilibration. We expect the latter scenario to be less likely, since we have observed a progressive reweighing of a bimodal isotopic distribution with none ($C_{12}BTA3D$) and all three amide groups deuterated ($C_{12}BTA6D$). When $D_2O$ penetrates the hydrophobic pocket, this would allow one amide group to undergo HDX at a time, resulting in a gradual shift of the isotopic distribution to higher masses. The HDX kinetics also clearly indicate that after 1 h of rapid exchange, there are two rates of HDX that differ one order of magnitude. Although there is no clear relation between these rate constants and the physical movement of the molecules, it is very surprising that even after three days of being dissolved in $D_2O$, over 40% of $C_{13}BTA3D$ is still present, while the other part is exchanged fully within a couple of hours. This suggests that our supramolecular polymers are structurally diverse. We hypothesize that some BTA monomers are weakly associated with surrounding BTAs, whereas other BTAs are more strongly associated and therefore exchange at a slower timescale. The circular dichroism spectra of the chiral BTA, strongly suggest the presence of a hydrogen-bonding helical arrangement of the supramolecular polymers studied here. This hydrogen bonding is most likely the reason why some of the exchange is so slow. We cannot completely rule out that polymerization–depolymerization at the ends of the BTA polymers contributes to the H/D exchange profile, but this process should not play a major role, as we reported using super resolution microscopy[34].

The kinetic data are reproducible, although we have observed that small changes in the rate constants $k_{slow}$ and $k_{fast}$ are sensitive to temperature and time of ageing. This observation calls for the importance of carefully monitoring and reporting the temperatures at which the samples are equilibrated before and after the dilution step. This pronounced sensitivity is likely due to the fact that with HDX-MS, all molecules present in solution are used for the readout. When a molecular probe (for example, spin label, dye) is employed, typically only a fraction of the molecules is labelled and therefore analysed. Previously, we reported similar

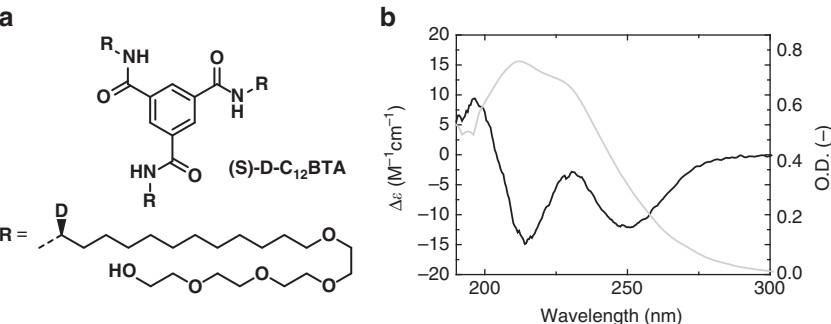

**Figure 6 | Chiral BTA assembly.** (a) Chemical structure of stereoselectively deuterated $(S)$-D-$C_{12}BTA$ and, (b) circular dichroism (black line) and UV- absorption spectrum (grey line) in water after equilibration for 4 weeks ($c = 50\,\mu M$, $T = 20\,°C$, path length = 1 cm).

kinetic profiles for $C_{11}BTA$ and $C_{12}BTA$ based on FRET experiments[38]. Here, we have observed clear differences in the HDX kinetics while gradually increasing the size of the aliphatic spacer of the molecules in steps of one $CH_2$ unit from $C_{10}BTA$ to $C_{13}BTA$. These results indicate that small changes in the molecular structure can be used to control the exchange dynamics of a synthetic supramolecular polymer, and confirm that HDX-MS is a suitable technique to obtain such kinetic information. Given the fact that the major advantage of a supramolecular polymer as compared to a covalent polymer is its dynamic nature, we anticipate that the ability to control the dynamics like we have shown here will be very important for future investigations.

In conclusion, we demonstrated that HDX-MS is a powerful technique for probing the dynamics of supramolecular polymers in water. Although it does not inherently provide structural data, HDX-MS can provide detailed information at a molecular level for synthetic supramolecular polymers without the limitations of introducing molecular labels. The different rate constants observed reveal that the BTA polymers are dynamically diverse; the ability of BTA monomers to move within or in and out of the supramolecular polymers is likely a continuously changing variant over the entire polymer in time. We anticipate that the observed dynamic diversity is not limited to the BTA polymers discussed here, and that HDX-MS will be widely applied to characterize many other supramolecular structures, such as bis-ureas, cyclohexyltrisamides, peptide amphiphiles and perylene bisimides[3–7].

## Methods

**General.** HDX-MS measurements were carried out using a Xevo G2 QTof mass spectrometer (Waters) with a capillary voltage of 2.7 kV and a cone voltage of 20 V. The source temperature was set at 100 °C, the desolvation temperature at 400 °C and the gas flow at $500 \, l \, h^{-1}$. The sample solutions subjected to HDX were introduced into the mass spectrometer using a Harvard syringe pump (11 Plus, Harvard Apparatus) at a flow rate of $50 \, \mu l \, min^{-1}$. $^1H$ NMR and $^{13}C$ NMR spectra were recorded on a Varian Mercury Vx 400 MHz (100 MHz for $^{13}C$), a Varian Mercury Plus 200 MHz (50 MHz for $^{13}C$), or a Bruker 400 MHz NMR spectrometer. Chemical shifts are given in p.p.m. (δ) values relative to residual solvent or tetramethylsilane. Splitting patterns are labelled as s, singlet; d, doublet; dd, double doublet; t, triplet; q, quartet; quin, quintet; m, multiplet and b stands for broad. Matrix-assisted laser desorption/ionisation mass spectra were obtained on a PerSeptive Biosystems Voyager DE-PRO spectrometer or a Bruker autoflex speed spectrometer using α-cyano-4-hydroxycinnamic acid (CHCA) and 2-[(2E)-3-(4-tert-butylphenyl)-2-methylprop-2-enylidene]malononitrile (DCTB) as matrices. Infrared spectra were recorded on a Perkin Elmer Spectrum One 1,600 FT-IR spectrometer or a Perkin Elmer Spectrum Two FT-IR spectrometer, equipped with a Perkin Elmer Universal ATR Sampler Accessory. CD measurements were performed on a Jasco J-815 spectropolarimeter where the sensitivity, time constant and scan rate were chosen appropriately. UV–vis absorbance spectra were recorded on a Jasco V-650 UV–vis spectrometer equipped with a Jasco ETCT-762 temperature controller.

**Synthesis.** The synthesis of $C_{12}BTA$ has previously been reported[31], as well as the synthesis of $C_{10}BTA$ and $C_{11}BTA$[38]. The synthetic methodologies for $C_{13}BTA$ and $(S)$-D-$C_{12}BTA$ are presented in detail in the Supplementary Methods. The full characterization of these compounds is presented in Supplementary Figs 4–12.

**Sample preparation.** The solid $C_{12}BTA$ was weighed into a glass vial. Deionized water was added to the glass vial in order to obtain a concentration of 500 μM in $H_2O$. The mixture was subsequently stirred, using a magnetic stir bar, at a temperature of 80 °C for 15 min. The hot and hazy mixture was vortexed for 15 s, and then left to equilibrate at room temperature overnight. The sample was diluted 100 times in $D_2O$ (including 0.5 mM sodium acetate to facilitate detection) using a balance, resulting in a concentration of 5 μM. The as-prepared BTA-solution was subsequently stored either in an oven for H/D exchange at 40, 50 or 55 °C for the duration of the experiment or at room temperature. Aliquots were periodically taken from the 5 μM solution and were subjected to ESI-MS. The sample preparation method for $C_{10}BTA$ and $C_{11}BTA$ samples was identical. Because of its reduced water-solubility, the sample preparation for $C_{13}BTA$ deviates. The solid $C_{13}BTA$ was weighed into a glass vial and deionized water was subsequently added to this glass vial to obtain a concentration of 50 μM in $H_2O$. The mixture was then stirred using a magnetic stir bar at a temperature of 80 °C for 15 min. The hot and

hazy mixture was vortexed for 15 s, and then left to equilibrate at room temperature overnight. After performing a UV-absorption measurement of the $C_{13}BTA$ sample, a significant amount of UV-absorption at wavelengths above 300 nm was observed. Since BTA-aggregates in water usually do not absorb light in this region, this is probably due to incomplete dissolution of the $C_{13}BTA$ aggregates. Therefore, the $C_{13}BTA$ sample was filtered using a 0.2 μm syringe filter (Supor membrane, PALL Corporation). Using a stream of $N_2$ (g), water was evaporated from the sample to increase the concentration for HDX-MS measurements. The final concentrations of the $C_{13}BTA$ samples prepared for the exchange experiments were determined using calibration curves based on the absorbance of $C_{12}BTA$ at 250 nm. The concentrations of the obtained $C_{13}BTA$ samples in $H_2O$, before the 100-fold dilution step, were 151, 276 and 453 μM. As compared to the other BTAs, the UV-absorption maximum at 209 nm is typically elevated relative to the UV-absorption maximum at 226 nm; a representative UV-spectrum is displayed in Supplementary Fig. 13.

**Fitting the exchange kinetics data.** The decrease in $C_{12}BTA3D$ over time was well-described with a tri-exponential fit. The fits were performed using the Origin 9.0.0 software package. In the Fitting Function Builder, the equation $y = A \times \exp(-t \times c) + B \times \exp(-t \times d) + E \times \exp(-t \times f)$ was used for the tri-exponential fits. Not taking into account the data points in the first hour, the equation $y = A \times \exp(-t \times c) + B \times \exp(-t \times d)$ was used for the bi-exponential fits. In these equations, A, c, B, d, E and f are the parameters that were varied during fitting and t was used as the independent variable (time in h).

**Data availability.** All data generated or analysed during this study are included in this published article (and its supplementary information files).

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

## Acknowledgements

We like to thank Jolanda Spiering for the synthetic support and Giovanni Pavan and Henk Janssen for the stimulating discussions. The work was financed by the Dutch Ministry of Education, Culture and Science (Gravity programme 024.001.035), the European Research Council (FP7/2007-2013, ERC Grant Agreement 246829).

## Author contributions

X.L. designed the HDX experiment; X.L., R.P.M.L. and S.M.C.S. carried out the HDX experiments; C.M.A.L., M.B.M. and N.M.M. synthesized the BTAs, and J.L.J.v.D., A.R.A.P. and E.W.M. contributed to the writing of the manuscript and the supervision of the research. R.P.M.L. wrote the paper with input from all authors.

## Additional information

**Competing interests:** The authors declare no competing financial interests.

