## [Peer Review File · Nature Communications]

Reviewers' comments:

Reviewer #1 (Remarks to the Author):

A driving force for generating supramolecular polymers is to mimic natural polymers, such as proteins. There, hydrogen bonds are the key for their self-assembly in solution. The senior author of this paper is a world known pioneer in supramolecular polymers exploiting multiple hydrogen bonds. In the present manuscript he and his co-workers apply hydrogen/deuterium exchange (HDX) mass spectrometry (MS), a technique well established in biochemistry to study the dynamic processes involved in synthetic supramolecular self-assemblies in water. This approach does not involve applications of spectroscopic labels, which have been used before. The experiments are very thorough and possible pitfalls have been extensively checked.

The results are rather intriguing as they indicate that the H/D exchange occurs on three different time scales and extends over exceeding long times, i.e. hours. This indicates that these supramolecular polymers in water are dynamically heterogeneous, well-known for synthetic polymers in bulk. These findings also create doubt on earlier studies of the dynamics of these polymers by the use of spectroscopic labels, which might distort these heterogeneities. Even more important is the finding that the differences between biomacromolecules and supramolecular polymers are larger than the similarities. As the major advantage of a supramolecular polymer as compared to a covalent polymer is its dynamic nature, the authors anticipate that the ability to control the dynamics as shown in this paper will be very important for future research in this field. Thus, the paper is of major interest for the broad readership of Nature Communications and it gives me pleasure recommending this fine work for publication.

Reviewer #2 (Remarks to the Author):

This manuscript illustrates that HDX-MS measurement can provide insight into the mechanism of molecular self-assembly. The authors group has so far made such effort to apply experimental methods used in biochemistry to characterize synthetic supramolecular systems: e.g., stopped flow in Nature 2012; STORM in Science 2014. This kind of translation has been indeed successful, and I believe this manuscript shows a new approach. Although this method allows only to assess the hydrogen-bonding exchange, the data obtained are not averaged but contains useful information on the inhomogeneity/diversity, thus are orthogonal to the data collected by other measurements (e.g. FRET study and IR spectroscopy). The experiments were performed carefully, starting with validation of this method and optimization of the conditions. Given the ever more increasing amount of attention paid to the mechanism of molecular self-assembly, this paper will add another approach unveiling the intricate mechanism of molecular self-assembly.

1) Figure 5 is quite interesting. It would be nice if k_{fast} and k_{slow} of the BTA derivatives with different alkyl chain lengths are compared.

2) As probed using CD spectra (Figure S4b), hydrogen-bonding becomes more organized over time. If time-dependent HDX-MS measurement is performed for samples having different degree of order (as a function of aging time), it would be possible to discuss the correlation between the hydrogen-bonding strength and H/D exchange kinetics (k_{fast} and k_{slow}), apart from the difference in the size of the hydrophobic pockets which is discussed in Figure 5.

What the authors found in this study (i.e., "diversity") could be specific phenomenon to BTA derivatives. Nevertheless, this paper will surely motivate other groups to investigate other systems similarly, by which further insights into the supramolecular self-assembly mechanism will be accumulated. What one should keep in mind to do so are also mentioned, which is helpful. As such, this paper is important in this research field, and I recommend the publication in Nature Communications.

Reviewer #3 (Remarks to the Author):

In this manuscript, the authors use HDX MS to illustrate diversity of dynamics of supramolecular polymers in water. This is illustrated with one class of polymers, benzene-1,3,5-tricarboxamide (BTA) derivatives that self-assemble in H₂O and have 6 exchangeable H (3 OH and 3 amide). The results are very clearly presented and do illustrate that small changes in structure (number of methylenes in alkyl chain) control the dynamics of the polymers. While this is useful information, the authors are unable to explain e.g., why after three days of being dissolved in D₂O, some 50% of C₁₃BTA₃D is still present along with 50% C₁₃BTA₆D (fully deuterated) - what structural feature of the polymer leads to this result. Although the authors have learned that there are primarily two rates of exchange of this type of polymer, they are unable to correlate that with any particular behavior or structure of the polymer. They also propose that their results suggest that HDX will be a useful tool, "equally important" to its use for proteins, for study of structures of supramolecular polymers in general. That is quite an extrapolation because they have illustrated this for only one structural type, with indirect structural information. The use of HDX as a tool for examination of protein structure is different in that it provides rich experimental data involving a variety of exchange rates over a large, complex folded polymer length and has been established even for membrane proteins in nanodiscs.

The authors have not fully answered the following questions:

Why is the use of HDX for monomer exchange characteristics (in supramolecular polymer structure) enough of an advance, over the huge amount of HDX work in the literature, to warrant publication in Nature Communications, with only one monomer type illustrated?

What are the variety of monomers used for supramolecular polymers, in general, and how many of them are likely to be amenable to detailed information from HDX? What types of information is the community missing that HDX might address?

When the following statement is made, it should be explained in more structural detail. "Some BTA monomers are weakly associated with surrounding BTAs, whereas other BTAs are more strongly associated and therefore exchange at a slower timescale." The authors know that change in the methylene chain length dramatically changes the protection of the amide H's. How is the hydrophobic pocket changing for the weakly associated and strongly associated monomers?

Minor comments:

The authors commented that they never saw dimers or lower oligomers and concluded that electrospray must depolymerize the sample. Protein chemists use very different spray conditions when they want to keep non-covalent assemblies intact (native MS conditions, nanospray and appropriate ionic strength). The authors might try nanospray ionization to produce oligomers.

When commenting on the following, it would be useful to state directly that you are suggesting a trace amount of back-exchange. "The tiny peak in front of C₁₂BTA₆D is due to the presence of a trace amount of the original H₂O."

Line 306, change built to build

Line 333, small spherical micelles are suggested without justification - should this be tied to data in supplemental?

Line 336, delete Although

Line 345, change part to 50%

We like to thank the reviewers for time and valuable comments. All points raised are taken into account and our answers (in blue) and our changes (in red) are given point-by-point below.

Point-by-point comments on the remarks of the reviewers.

Reviewer #1 (Remarks to the Author):

A driving force for generating supramolecular polymers is to mimic natural polymers, such as proteins. There, hydrogen bonds are the key for their self-assembly in solution. The senior author of this paper is a world known pioneer in supramolecular polymers exploiting multiple hydrogen bonds. In the present manuscript he and his co-workers apply hydrogen/deuterium exchange (HDX) mass spectrometry (MS), a technique well established in biochemistry to study the dynamic processes involved in synthetic supramolecular self-assemblies in water. This approach does not involve applications of spectroscopic labels, which have been used before. The experiments are very thorough and possible pitfalls have been extensively checked.

The results are rather intriguing as they indicate that the H/D exchange occurs on three different time scales and extends over exceeding long times, i.e. hours. This indicates that these supramolecular polymers in water are dynamically heterogeneous, well-known for synthetic polymers in bulk. These findings also create doubt on earlier studies of the dynamics of these polymers by the use of spectroscopic labels, which might distort these heterogeneities. Even more important is the finding that the differences between biomacromolecules and supramolecular polymers are larger than the similarities. As the major advantage of a supramolecular polymer as compared to a covalent polymer is its dynamic nature, the authors anticipate that the ability to control the dynamics as shown in this paper will be very important for future research in this field. Thus, the paper is of major interest for the broad readership of Nature Communications and it gives me pleasure recommending this fine work for publication.

We thank the Reviewer for critically reading our manuscript and sharing his/her enthusiasm about our work with us. It is a pleasure to read a very accurate description that explains the importance of the presented results.

Reviewer #2 (Remarks to the Author):

This manuscript illustrates that HDX-MS measurement can provide insight into the mechanism of molecular self-assembly. The authors group has so far made such effort to apply experimental methods used in biochemistry to characterize synthetic supramolecular systems: e.g., stopped flow in Nature 2012; STORM in Science 2014. This kind of translation has been indeed successful, and I believe this manuscript shows a new approach. Although this method allows only to assess the hydrogen-bonding exchange, the data obtained are not averaged but contains useful information on the inhomogeneity/diversity, thus are orthogonal to the data collected by other measurements (e.g. FRET study and IR spectroscopy). The experiments were performed carefully, starting with validation of this method and optimization of the conditions. Given the ever more increasing amount of attention paid to the mechanism of molecular self-assembly, this paper will add another approach unveiling the intricate mechanism of molecular self-assembly.

We thank the Reviewer for critically reading our manuscript and sharing his/her enthusiasm about the work with us.

1) Figure 5 is quite interesting. It would be nice if k_{fast} and k_{slow} of the BTA derivatives with different alkyl chain lengths are compared.

We thank the Reviewer for this valuable comment. We have been hesitant to interpret all the rate constant k_{fast} because of their relative contributions and the small difference between the initial and fast part; therefore the data differ significantly (see supplementary methods part 4). With confidence, only k_{slow} can be compared. This can also be observed qualitatively by looking at Figure 5b, so we agree with the Reviewer that this comparison is a nice addition. Therefore, we added the following sentence to the manuscript (line 278). *“These differences in the conversion to the fully deuterated species are reflected in the rate constants k_{slow} that are $1.5 \times 10^{-2} \text{ h}^{-1}$, $0.9 \times 10^{-2} \text{ h}^{-1}$ and $0.3 \times 10^{-2} \text{ h}^{-1}$ for increasing the size of the aliphatic spacer, respectively.”*

2) As probed using CD spectra (Figure S4b), hydrogen-bonding becomes more organized over time. If time-dependent HDX-MS measurement is performed for samples having different degree of order (as a function of aging time), it would be possible to discuss the correlation between the hydrogen-bonding strength and H/D exchange kinetics (k_{fast} and k_{slow}), apart from the difference in the size of the hydrophobic pockets which is discussed in Figure 5.

We fully agree with the Reviewer that it would have been interesting to perform HDX-MS measurements with the chiral molecule. Unfortunately we ran out of the (*S*)-D-C₁₂BTA. Since we did not expect large differences in the rate constants to delineate the hydrophobic and hydrogen bonding effects with certainty, we did not restart the synthesis of (*S*)-D-C₁₂BTA, also because this molecule takes several months to synthesize.

3) What the authors found in this study (i.e., “diversity”) could be specific phenomenon to BTA derivatives. Nevertheless, this paper will surely motivate other groups to investigate other systems similarly, by which further insights into the supramolecular self-assembly mechanism will be accumulated. What one should keep in mind to do so are also mentioned, which is helpful. As such, this paper is important in this research field, and I recommend the publication in Nature Communications.

We thank the Reviewer for these compliments.

Reviewer #3 (Remarks to the Author):

In this manuscript, the authors use HDX MS to illustrate diversity of dynamics of supramolecular polymers in water. This is illustrated with one class of polymers, benzene-1,3,5-tricarboxamide (BTA) derivatives that self-assemble in H₂O and have 6 exchangeable H (3 OH and 3 amide). The results are very clearly presented and do illustrate that small changes in structure (number of methylenes in alkyl chain) control the dynamics of the polymers. While this is useful information, the authors are unable to explain e.g., why after three days of being dissolved in D₂O, some 50% of C₁₃BTA₃D is still present along with 50% C₁₃BTA₆D (fully deuterated) - what structural feature of the polymer leads to this result. Although the authors have learned that there are primarily two rates of exchange of this type of polymer, they are unable to correlate that with any particular behavior or structure of the polymer. They also propose that their results suggest that HDX will be a useful tool, “equally important” to its use for proteins, for study of structures of supramolecular polymers in general. That is quite an extrapolation because they have illustrated this for only one structural type, with indirect structural information. The use of HDX as a tool for examination of protein structure is different in that it provides rich experimental data involving a variety of exchange rates over a large, complex folded polymer length and has been established even for membrane proteins in nanodiscs.

We agree with the Reviewer that this is an overstatement; we were too enthusiastic and have removed “equally important” in the last line (line 378). We thank the Reviewer for the nice comments on the supramolecular aspects.

The authors have not fully answered the following questions:

- 1) Why is the use of HDX for monomer exchange characteristics (in supramolecular polymer structure) enough of an advance, over the huge amount of HDX work in the literature, to warrant publication in Nature Communications, with only one monomer type illustrated?

Currently, there are no experimental techniques available that can probe supramolecular polymers at both the temporal and spatial resolution that would be required to relate the dynamics to structural features and hence to their properties. The precise packing of the molecules inside supramolecular polymers in water still requires extensive characterization; even though the field is rapidly progressing, predicting the relation between the molecular structure of the monomer and their packing inside these dynamic aggregates is still a “holy grail” in supramolecular chemistry. As these polymers are foreseen to be used as important building blocks for the next generation of biomaterials, detailed analysis is a prerequisite.

The advantage of supramolecular polymers as compared to covalent synthetic polymers is their dynamic nature. To study the kinetics of supramolecular monomers, early work has been performed by attaching molecular probes like spin labels (10.1038/nmat3979) or dyes (10.1126/science.1250945). In recent years it has become apparent that small changes in molecular structure do have a large influence on the kinetic properties of these polymers. Therefore, the use of HDX to study kinetics, without a sterically bulky molecular probe, is a major advance in the field.

Although HDX MS is well-known in biochemistry and biophysics, this technique is new to supramolecular chemists. Obviously, the HDX processes for the supramolecular self-assemblies and for proteins share many similarities, but they also have some significant differences. For proteins with three dimensional structures, HDX events are generally mediated by conformational fluctuations with open/close conformations. However, our BTA self-assemblies are one dimensional and are held together only by supramolecular interactions. By breaking down the supramolecular interactions, a monomer will be released from the supramolecular polymer. In contrast, the primary structure of a protein will not change during the HDX process since the amino acids are linked together via covalent bonds. From these perspectives, it appears that the differences between proteins and synthetic supramolecular polymers in HDX and in the subsequent MS analysis can be significant.

We have selected the C₁₂BTA motif to assess the applicability of HDX-MS for studying supramolecular polymers because this is the main building block we have been investigating in recent years for self-assembly in water because of its promising applications as biocompatible hydrogel and in signal transduction.

Moreover, control over the ‘non-covalent synthesis’, i.e. how to reproducibly prepare similar samples of supramolecular polymers in water, is not trivial and we carefully stick to the sample preparation protocols as described. Like the BTA, aggregates formed from different monomers require different delicate sample preparation procedures and optimizing these, together with the optimization of the HDX-MS experiment, is beyond the scope of the current work. We envisage that the intriguing presence of multiple timescales will also encourage other supramolecular chemistry groups to apply this powerful technique.

- 2) What are the variety of monomers used for supramolecular polymers, in general, and how many of them are likely to be amenable to detailed information from HDX? What types of information is the community missing that HDX might address?

There are a plethora of monomers currently being studied and the amount is ever increasing. A large variety of supramolecular polymers in water have recently been reviewed (10.1021/acs.chemrev.5b00369). In our introduction we have also cited a series of well-known building blocks among which the bis-ureas, trisamide-cyclohexanes, peptide-amphiphiles and perylene bisimides. We expect that all these are amenable to detailed information from HDX-MS. Therefore we changed the last sentence of the manuscript (line 378). *“We anticipate that the observed dynamic diversity is not limited to the BTA polymers discussed here, and that HDX-MS will be widely applied to characterize many other supramolecular structures, such as bis-ureas, cyclohexyltrisamides, peptide amphiphiles and perylene bisimides³⁻⁷.”*

It is difficult to speculate on the discoveries – to be made – when other supramolecular groups will start to use the technique. For the BTAs, the kinetic behaviour we have observed with HDX-MS was previously completely missed when ensemble techniques such as FRET (10.1039/C5SM02843D) and STORM (10.1126/science.1250945) were used. Although some beautiful work illustrating how small changes in molecular structure can influence the aggregation of supramolecular building blocks in organic solvents have recently been published (e.g. 10.1021/jacs.5b11674 and 10.1038/nchem.2684), the translation to an aqueous environment is always challenging. What the community of supramolecular chemistry is missing, is to obtain the kinetic information using a truly label-free technique. We therefore expect that there are many opportunities ahead for supramolecular chemists to exploit the unique capabilities of HDX-MS.

- 3) When the following statement is made, it should be explained in more structural detail. "Some BTA monomers are weakly associated with surrounding BTAs, whereas other BTAs are more strongly associated and therefore exchange at a slower timescale." The authors know that change in the methylene chain length dramatically changes the protection of the amide H's. How is the hydrophobic pocket changing for the weakly associated and strongly associated monomers?

We recognize the opinion of the Reviewer that this particular sentence requires an explanation with more structural detail. In our answer to the second remark of this Reviewer we have explained that we lack enough detailed structural information to correlate with the kinetic data. This statement should have been an assumption, and therefore we changed this sentence in the new version of the manuscript (line 346). *“We hypothesize that some BTA monomers are weakly associated with surrounding BTAs, whereas other BTAs are more strongly associated and therefore exchange at a slower timescale.”* Possibly, some BTAs stick to the outer side of the polymer and are ‘weakly associated’, whereas others are fully intercalated in the polymer and are therefore better protected from surrounding water by the hydrophobic effect (‘strongly associated’). We would love to have a rational based on experiments, but currently we can only speculate on these nanometer-structural details.

Minor comments:

The authors commented that they never saw dimers or lower oligomers and concluded that electrospray must depolymerize the sample. Protein chemists use very different spray conditions when they want to keep non-covalent assemblies intact (native MS conditions, nanospray and appropriate ionic strength). The authors might try nanospray ionization to produce oligomers.

We fully agree with the Reviewer that non-covalent assemblies might be observed using different spray conditions. However, the aim of this study is not to measure intact assemblies, and the suggestion may be interesting for future work. The absence of dimers or lower oligomers indeed indicates that quick depolymerization occurs during the ionization process using the particular conditions we employed. In order to better communicate that this is a consequence of the spray conditions, we changed the sentence (in line 149) in the new version of the manuscript accordingly. *“We never observed masses corresponding to dimers or lower oligomers in the MS spectra, indicating that full depolymerisation of the supramolecular polymers occurred as a consequence of the ionization settings we employed.”*

When commenting on the following, it would be useful to state directly that you are suggesting a trace amount of back-exchange. "The tiny peak in front of C₁₂BTA6D is due to the presence of a trace amount of the original H₂O."

We respectfully disagree with this comment. Unlike HDX-MS for protein samples, no digestion and subsequent LC separation is required for the supramolecular polymers. The sample solution after HDX is directly introduced into the ion source. Therefore, the tiny peak in front of C₁₂BTA6D is not caused by back-exchange. It is due to the presence of a trace amount of the original H₂O.

Line 306, change built to build

We thank the Reviewer for pointing out this typing error.

Line 333, small spherical micelles are suggested without justification - should this be tied to data in supplemental?

We thank the Reviewer for pointing this out and made the following changes to lines 332-336. *“We propose three possible reasons for the very fast exchange observed in the first hour. Firstly, the very fast exchange could be due to the presence of small micelles that are not stabilized by intermolecular hydrogen bonds. Secondly, the 100-fold dilution step might shed weakly incorporated monomers from the polymer, or temporarily allow the penetration of D₂O into the hydrophobic pocket during re-equilibration.”*

Line 336, delete Although.

We have deleted Although.

Line 345, change part to 50%.

We discovered a typing error in the line before, so ‘part’ is now correct in the new version of the manuscript.

REVIEWERS' COMMENTS:

Reviewer #2 (Remarks to the Author):

The authors sincerely replied to the comments raised by these three reviewers, and this manuscript is publishable in Nature Communications. I believe that this manuscript will motivate the researchers in this field to pursue the complex behaviors, which is surely beneficial to advance synthetic supramolecular systems.

Reviewer #3 (Remarks to the Author):

While I believe that some readers will question the novelty of this work (HDX is a well-established tool), others will appreciate the illustration of this new application. The authors have made some useful changes. I disagree with their statement about residual water - back-exchange can occur in the spray process independent of the use of LC, depending on experimental conditions - but this is a minor point. I recommend publication.

Answer to Reviewers

Response to the remarks of the 2nd reviewer.

We thank the reviewer for the encouragement.

Response to the remarks of the 3rd reviewer.

We thank the reviewer for the remarks and his/her recommendation for publication.

We fully agree with the reviewer that back-exchange can also occur in the spray process if the ratio of D_2O/H_2O is decreased. In our case, no extra H_2O was introduced and the ratio of D_2O/H_2O was remained at 100/1 throughout the experiments. Because of the deuterium rich environment, some additional H/D exchange, instead of back-exchange, might occur during the spray process. Fortunately, we found that the contribution of H/D exchange in the spray process is negligible for our synthetic supramolecular polymers under our experimental conditions (see the section of “validation of HDX-MS”).